# Antibacterial Metabolites Produced by *Limonium lopadusanum,* an Endemic Plant of Lampedusa Island

**DOI:** 10.3390/biom14010134

**Published:** 2024-01-22

**Authors:** Ernesto Gargiulo, Emanuela Roscetto, Umberto Galdiero, Giuseppe Surico, Maria Rosaria Catania, Antonio Evidente, Orazio Taglialatela-Scafati

**Affiliations:** 1Department of Pharmacy, University of Naples Federico II, Via Domenico Montesano, 49, 80131 Napoli, Italy; ernesto.gargiulo@unina.it; 2Department of Molecular Medicine and Medical Biotechnologies, University of Naples Federico II, Via Pansini 5, 80131 Napoli, Italy; emanuela.roscetto@unina.it (E.R.); umberto.galdiero@unina.it (U.G.); mariarosaria.catania@unina.it (M.R.C.); 3Department of Agriculture, Food, Environment, and Forestry (DAGRI), Section of Agricultural Microbiology, Plant Pathology and Entomology, University of Florence, 50121 Firenze, Italy; giuseppe.surico@unifi.it; 4Institute of Sciences of Food Production, National Research Council, Via Amendola 122/O, 70125 Bari, Italy; evidente@unina.it

**Keywords:** *Limonium lopadusanum*, biological activities, phenylpropanoids, antibiotic activity

## Abstract

Lampedusa, the largest island of the Pelagie archipelago, Sicily, Italy, has proven to be a rich source of plants and shrubs used in folk medicine. These plants, often native to the island, have been very poorly investigated for their phytochemical composition and biological potential to be translated into pharmacological applications. To start achieving this purpose, a specimen of *Limonium lopadusanum*, a plant native to Lampedusa, was investigated for the first time. This manuscript reports the results of a preliminary biological assay, focused on antimicrobial activity, carried out using the plant organic extracts, and the isolation and chemical and biological characterization of the secondary metabolites obtained. Thus 3-hydroxy-4-methoxybenzoic acid methyl ester (*syn*: methyl isovanillate, (**1**), methyl syringate (**2**), pinoresinol (**3**), erythrinassinate C (**4**) and tyrosol palmitate (**5**) were isolated. Their antimicrobial activity was tested on several strains and compound **4** showed promising antibacterial activity against *Enterococcus faecalis*. Thus, this metabolite has antibiotic potential against the drug-resistant opportunistic pathogen *E. faecalis.*

## 1. Introduction

The requests for new, effective, and eco-friendly tools to deal with environmental changes, the increasing food need and emerging human diseases have become urgent. Natural products are still an unmatched source of compounds with different biological activities and new carbon skeletons, and thus, they represent important means for agriculture, medicine, and several other fields. Plants are a rich reservoir of secondary metabolites that can be used as defense, hormones or to attract pollinators, with potential applications in food chemistry, agriculture, and medicine [1]. The naturally occurring compounds have a potential to overcome the resistance phenomena developed by invasive plants, dangerous insects, and pathogenic microorganisms. In this context, the investigation of poorly studied endemic plants holds a great potential to unveil unprecedented chemistry and/or biology.

Lampedusa, the largest island of the Pelagie archipelago, off the coasts of Sicily, Italy, has proven to be a rich source of plants and shrubs used in folk medicine. These plants, often native to the island, have been little-studied or not studied at all for the chemical and biological characterization of their secondary metabolites and the evaluation of their potential applications, especially in medicine and agriculture. Thus, a preliminary screening was carried out for several Lampedusa endemic plant species, showing that many organisms hold promising biopesticide, antiviral, and antibacterial properties [2]. For example, *Thymelaea hirsuta* produced a variety of secondary metabolites, such as chromenes, cyclohexanones, furanyl, bis-furanyl and furanone polyphenols, and acrylates. Tested against three *Colletotrichum* species (*C. acutatum*, *C. fragarie* and *C. gloeosporioides*), pathogenic fungi for agricultural plants, 6-hydroxy-4,4,7a-trimethyl-5,6,7,7a-tetrahydro-4*H*-benzofuran-2-one showed activity against all three species, although it was not as active as Captan, the commercial fungicide used as a positive control [3].

*Limonium lopadusanum* Brullo (Plumbaginaceae) samples were found during the investigation on the impact of *Pinus halepensis* plantations under a semi-arid climate, to evaluate the combined effect of soil treatment and afforestation practices on spontaneous plant species compositions of Lampedusa [4]. The genus name *Limonium* derives from the Greek “λειμών” which means humid and grassy place, meadow, while the species name *lopadusanum* comes from the name of the island Lopadusa, Latin translation of the Greek word λοπχδοϋσσχ (Lopadoussa). *L. lopadusanum* is a plant up to 20-30 cm tall, has cushion rosettes and small obovate-spatulate leaves (Figure 1). It flowers between June and August and grows abundantly on the limestone coastal cliffs of the island, but it is also present in several locations of the main island Sicily. 

The genus *Limonium* is well known to produce bioactive secondary metabolites as the antioxidants myricetin 3-O-α-rhamnopyranoside, (-)-epigallocatechin 3-*O*-gallate, (-)-epigallocatechin, (+)-gallocatechin, and gallic acid [5]. 5-Methylmyricetin, myricetin, and myricetin-3-*O*-β-glucoside showed good antifungal activity against *Candida glabrata*, while only myricetin exhibited significant antimalarial activity against resistant and sensitive strains of *Plasmodium falciparum* [6]. (2*R*,3*S*)-2,3,4-trihydroxy-2-methylbutyl gallate, isolated from *Limonium leptophyllum,* also showed good activity against *P. falciparum. Limonium morisianum* methanolic extract showed dual inhibition of HIV-1 reverse transcriptase (RT) and integrase (IN), and its metabolites (-)-epigallocatechin-3-*O*-gallate and myricetin-3-*O*-(6″-*O*-galloyl)-β-D-galactopyranoside strongly inhibited both enzymes [7].

On these bases and considering that, until now, there is no literature report on the secondary metabolites of *L. lopadusanum* and on their biological properties, this manuscript describes the results of preliminary antimicrobial tests against important clinical pathogens performed using the plant organic extract, the first identification of metabolites and their antibacterial activity. 

## 2. Materials and Methods

### 2.1. General Experimental Procedure

^1^H and ^13^C NMR spectra were recorded at 400 and 100 MHz, respectively, in CDCl_3_ on a Bruker (Billerica, MA, USA) spectrometer. The same solvent was used as an internal standard. Carbon multiplicities were determined by DEPT spectra. DEPT, COSY-45, HSQC, HMBC, and NOESY experiments were performed using Bruker microprograms. HRESIMS experiments were performed on an LTQ-Orbitrap Thermo Fisher (Waltham, MA, USA) mass spectrometer equipped with an electrospray (ESI) interface and Excalibur data system. Analytical and preparative TLC were performed on silica gel plates (Kieselgel 60, F254, 0.25 and 0.5 mm, respectively) or on reverse phase (KC18 F254, 0.20 mm) plates and the compounds were visualized by exposure to UV light and/or iodine vapors and/or by spraying first with 10% H_2_SO_4_ in MeOH, and then with 5% phosphomolybdic acid in EtOH, followed by heating at 110 °C for 10 min. CC: silica gel Merck (Darmstad, Germany) Kieselgel 60, 0.063–0.200 mm. HPLC-RI separations were performed on Knauer (Berlin, Germany) instruments, using Knauer 1800 apparatus equipped with a refractive index detector using Luna 5 μm Silica (2) 100 Å 250 × 4.6 mm column equipped with a Rheodyne injector.

### 2.2. Plant Material 

Whole aerial parts of *L. lopadusanum* plant samples were collected fresh on 14–19 April 2022 in different sites of the northern coast of Lampedusa Island (Italy) by Fabio Giovanetti and identified by one of the authors (G.S.). A voucher specimen is deposited in the collection of the Department of Agriculture, Food, Environment, and Forestry (DAGRI), Section of Agricultural Microbiology, Plant Pathology and Entomology, University of Florence, Italy, voucher n. DAGRI 56.

### 2.3. Extraction 

Fresh plant materials (1.5 kg) were extracted (1 × 3.0 L) with H_2_O/MeOH (1/1, *v*/*v*) under stirred conditions at room temperature for 24 h, and the obtained suspensions centrifuged at 7000 rpm at 5 °C. Supernatants were separated from the solid phase and extracted using *n*-hexane (3 × 1.0 L). The organic extracts were combined, dried (Na_2_SO_4_), filtered and evaporated under reduced pressure, yielding an oily residue (60.1 mg). The residual aqueous phase was extracted with CH_2_Cl_2_ (3 × 1.0 L). The organic extracts were combined, dried (N_2_SO_4_), filtered and evaporated under reduced pressure, yielding an oily residue (347.6 mg). The residual plant material of the first extraction was subjected to a second extraction giving a *n*-hexane extract (47.9 mg) and a CH_2_Cl_2_ extract (173.4 mg). Both *n*-hexane and the CH_2_Cl_2_ extracts were combined giving a total oily residue of 108.0 and 521.0 mg, respectively, which were assayed for their antibiotic activity. 

### 2.4. Purification of Metabolites from the L. lopadusanum Extracts

The crude bioactive CH_2_Cl_2_ extract (521.0 mg) obtained using the procedure described in Section 2.3 was purified by a silica gel column, eluted with CHCl_3_-*iso*-PrOH (95:5) and seven homogeneous fractions were collected (F1–F7) and monitored by TLC using the same solvent. Then, the column was washed with MeOH. The oily residue of fraction F5 and F6 were combined (149.7 mg) and an aliquot (60 mg) was further purified by preparative TLC, eluted with *n*-hexane-EtOAc (65:35) yielding seven fractions (LL21A–LL21F). The residue (17.9) of fraction LL21C appeared to be a homogeneous compound identified as (below reported) 3-hydroxy-4-methoxybenzoic acid methyl ester (syn: methyl isovanillate) (**1**, 17.9 mg). Thus, the residual part of the F5–F6 combined fraction was purified using the same procedure affording six fractions (LL25A–LL25F). The residue of fraction LL25C (13.1 mg) afforded an additional amount of compound **1**, for a total of 31.0 mg. The residue of fraction LL21F and LL25F were combined (97.0 mg) and purified by preparative TLC, eluted with *n*-hexane –EtOAc, (6:4) yielding five fractions (LL26A–LL26E). The residue of fraction LL26B (4.4 mg) was a homogeneous compound identified as syringic acid methyl ester (**2**). The residue of fraction LL25D (13.3 mg) was further purified using the same conditions giving four fractions (LL28A–LL28D). The residue of fraction LL28B (5.5 mg) is a further amount of compound **2** for a total of 9.9 mg. The residue of fraction LL26G (14.0 mg) was further purified by preparative TLC affording eight fractions (LL29A–LL29H). The residue of fraction LL29A appeared to be a homogeneous compound and was identified as (below reported) pinoresinol (**3**). The residue of fraction LL29B was purified by HPLC on silica gel column, eluent *n*-hexane/EtOAc 9:1, flow 1.0 mL/min to obtain pure erythrinassinate C (**4,** 2.1 mg) and tyrosol palmitate (**5**, 0.6 mg). 

#### 2.4.1. 3-Hydroxy-4-methoxybenzoic Acid Methyl Ester (**1**)

Compound **1**: HRESI-MS, *m/z* 205.0480 (calc. for C_9_H_10_O_4_Na *m/z* 205.0477). ^1^H NMR, δ: 7.64 (1H, dd, *J* = 8.2 and 1.9 Hz, H-6), 7.55 (1H, d, *J* = 1.9 Hz, H-2), 6.93 (1H, d, *J* = 8.2 Hz, H-5), 3.94 (3H, s, OCH_3_) 3.89 (3H, s, CO_2_CH_3_); ^13^C NMR, δ: 166.9 (CO_2_CH_3_), 150.0 (C-3), 146.2 (C-4), 124.2 (C-6), 122.3 (C-1), 114.1 ( C-5), 111.7 (C-2), 56.1(OCH_3_), 52.0 (CO_2_CH_3_).

#### 2.4.2. Syringic Acid Methyl Ester (**2**)

Compound **2**: HRESI-MS, *m/z* 235.0580 (calc. for C_10_H_12_O_5_Na *m/z* 235.0582). ^1^H NMR, δ: 7.32 (2H, s, H-2 and H-6), 3.94 (6H, 2xOCH_3_), 3.90 (3H, CH_3_); ^13^C NMR, δ: 166.7 (CO_2_Me), 146.0 (C-3, C-5), 136.1 (C-4), 121.0 (C-1), 106.3 (C-2, C-6), 56.4 (2xOCH_3_), 51.9 (CH_3_). 

#### 2.4.3. Pinoresinol (**3**)

Compound **3**: HRESI-MS, *m/z* 235.0580 (calc. for C_10_H_12_O_5_Na *m/z* 235.0582). ^1^H NMR, δ: 7.32 (2H, s, H-2 and H-6), 3.94 (6H, 2xOCH_3_), 3.90 (3H, CH_3_); ^13^C NMR, δ: 166.7 (CO_2_Me), 146.0 (C-3, C-5), 136.1 (C-4), 121.0 (C-1), 106.3 (C-2, C-6), 56.4 (2xOCH_3_), 51.9 (CH_3_). 

#### 2.4.4. Erythrinassinate C (**4**)

Compound **4**: HRESI-MS, *m/z* 413.2670 (calc. for C_24_H_38_O_4_Na *m/z* 413.2668). ^1^H NMR, δ: 7.60 (1H, d, *J* = 16.0 Hz, H-7); 7.07 (1H, d, *J* = 8.6 Hz, H-6), 7.03 (1H, bs, H-2), 6.90 (1H, d, *J* = 8.6 Hz, H-5), 6.26 (1H, d, *J* = 16.0 Hz, H-8), 4.19 (2H, t, *J* = 7.3 Hz, H-1′), 3.92 (3H, s, 3-OMe), 1.10–1.39 [24H, overlapped, H-2′ to H-13′], 0.88 (3H, t, *J* = 7.3 Hz, H-14′). ^13^C NMR: δ: 14.1 (C-14′), 22.7 (C-13′), 26.0 (C-3′), 28.8 (C-2′), 29.3 (C-4′ and C-11′), 29.7 (C-5′ to C-10′), 31.9 (C-12′), 55.9 (3-OMe), 64.6 (C-1″), 109.3 (C-2), 114.7 (C-5), 115.7 (C-8), 123.0 (C-6), 127.1 (C-1), 144.6 (C-7), 147.9 (C-3, C-4), 167.3 (C-9).

#### 2.4.5. Tyrosol Palmitate (**5**)

Compound **5**: HRESI-MS, *m/z* 399.2870 (calc. for C_24_H_40_O_3_Na *m/z* 399.2875). ^1^H NMR, δ: 7.07 (2H, d, *J* = 8.6 Hz, H-2 = H-6), 6.77 (2H, d, *J* = 8.6 Hz, H-3 =H-5), 4.24 (2H, t, *J* = 7.2 Hz, H-8), 2.85 (2H, t, *J* = 7.2 Hz, H-7), 2.27 (2H, t, *J* = 7.2 Hz, H-2′), 1.10–1.63 [26H, overlapped, H-3′ to H-15′], 0.88 (3H, t, *J* = 7.3 Hz, H-16′). ^13^C NMR: δ: 14.1 (C-16′), 22.7 (C-15′), 34.4 (C-7), 64.9 (C-8), 116.1 (C-3 = C-5), 129.9 (C-2 = C-6), 130.7 (C-1), 154.1 (C-4), 173.9 (C-1′).

### 2.5. Antimicrobial Assay 

The antimicrobial activity of *L. lopadusanum* extracts was assayed on bacterial and fungal reference strains: two Gram-positive bacteria such as methicillin-resistant *Staphylococcus aureus* (MRSA) ATCC 43300 and *Enterococcus faecalis* ATCC 29212, three Gram-negative bacteria such as *Escherichia coli* ATCC 25922, *Acinetobacter baumannii* BAA747 and *Pseudomonas aeruginosa* ATCC 27853, and the yeast voriconazole-resistant *Candida albicans* 10231. The antibacterial effect of secondary metabolites was assayed against *Enterococcus faecalis*. All strains were stored as 15% (*v/v*) glycerol stocks at −80 °C. The identification was carried out by MS MALDI-TOF Bruker (Bremen, Germany) and their antibiotic susceptibility profiles were evaluated by Vitek 2 (bioMérieux, Marcy-l’Étoile, France). The antimicrobial tests were performed through standard broth-microdilution assay in 96 well microtiter plates using brain heart infusion (BHI) broth. Starting from microbial suspensions with a turbidity of 0.5 McFarland for each test strain, the inoculum was diluted 1:100 in BHI-broth and a volume of 100 μL of the inoculum was added to each well. The antimicrobial activity was tested by adding 100 μL of serial dilutions (1:2) of test substances to the wells starting from 1000 μg/mL for plant extract and 500 μg/mL for purified metabolites. The wells without substances were used as a positive growth control. Teicoplanin (from 0.06 to 4 μg/mL) and colistin (from 0.5 to 32 μg/mL) were used as control for Gram-positive and Gram-negative, respectively; voriconazole (from 15 to 60 μg/mL) was used as conventional control for *Candida* strain. Serial dilutions of compound solvent (starting from 1% DMSO) were tested to be sure that it did not act on the bacterial growth. Thereafter, the plates were incubated at 37 °C under aerobic conditions for 20 h. The minimal inhibitory concentration (MIC) and minimal bactericidal concentration (MBC) of test substances were determined: the MIC was defined as the lowest concentration of substance that caused no visible bacterial growth in the wells. The MBC was defined as the lowest concentration of substance that kills the cells in the planktonic culture. The extracts and each compound were tested in triplicate and each experiment was performed twice.

## 3. Results and Discussion

### 3.1. Extraction and Preliminary Biological Assays

Specimens of *L. lopadusanum* were collected in Lampedusa Island and preliminarily extracted with different solvents with increasing polarity. The most efficient procedure resulted in extraction of the plant with *n*-hexane and CH*_2_*Cl_2_ in succession. The antimicrobial activity was preliminarily evaluated for the two extracts, starting from 1 mg/mL (dilutions 1:2) against *Staphylococcus aureus* (MRSA), *Escherichia coli*, *Pseudomonas aeruginosa*, *Enterococcus faecalis*, *Acinetobacter baumanii*, and the yeast *Candida albicans*. *n*-Hexane extract had no activity on these strains, while the CH_2_Cl_2_ extract exhibited a selective antibacterial activity against *E. faecalis* with a MIC value of 500 µg/mL (Table 1). 

Thus, the CH_2_Cl_2_ organic extract obtained from fresh aerial parts of *L. lopadusanum* was purified by a combination of CC, TLC and HPLC, as reported in detail in the Section 2. Five metabolites (**1**–**5**, Figure 2) were obtained in the pure form and their structures were elucidated by a detailed MS and NMR investigation (see Appendix A) and by comparison with data reported in the literature. Compound **1** was identified as 3-hydroxy-4-methoxybenzoic acid methyl ester (metyhl isovanillate) by comparing its ^1^HNMR data with those reported in the literature when it was isolated as a mild anti-inflammatory and analgesic agent from *Glycosmis pentaphylla* leaves, a plant used in Bangladesh folk medicine to treat different diseases [8]. Further supports to the structure assigned to **1** were successively obtained comparing its ^1^H and ^13^C NMR data with those recorded in acetone-d_6_ when it was isolated from *Dianthus superbus* L. (“Qu Mai” in Chinese), a plant widely distributed in China [9]. Compound **1** and some analogues can be easily oxidized to the corresponding cyclohexa-2,4-dienones, which are a synthetically useful class of compounds. In fact, 1,2 masked *o*-benzoquinones (MOBs) can undergo highly regio- and stereo-selective Diels-Alder reactions with electron-rich dienophiles affording interesting bicyclo[2.2.2]octenones [10]. The complete NMR assignment for compound **1**, reported in the Section 2, was obtained by ESIMS and 2D NMR data. Interestingly, isovanillic acid (3-hydroxy-4-methoxybenzoic acid) was isolated from *Treculia obovoidea*, which is distributed in the humid regions of Africa, from Nigeria to Congo and used in traditional medicine to treat skin diseases, dental allergy, amoebic dysentery, and AIDS [11].

Compound **2** was identified as the methyl ester of syringic acid by analyzing its ^1^H and ^13^C NMR spectra, which appeared consistent with a symmetric aromatic tetra-substituted benzene. In particular, its ^1^H NMR spectrum showed only the presence of three singlets, a 2H-integrating signal at δ 7.59, a 6H-integrating signal at δ 3.94, due to two methoxy groups, and a 3H-integrating signal at δ 3.90 due to an additional methoxy group. The ^13^C NMR spectrum showed the signal of an ester carbonyl at δ 166.4, the quaternary oxygenated aromatic carbons at δ 146.0 (C-3, C-5) and 136.1 (C-4) and of the protonated aromatic carbon at δ 106.3 (C-3, C-6). Full NMR assignment and structure confirmation was obtained by inspection of 2D NMR spectra (COSY, HSQC and HMBC) and comparison with the literature data [12]. Interestingly, methyl syringate was found to possess a mild antibacterial activity against *S. aureus* [12], lower than that detected for the corresponding free acid. It could be noted that syringic acid is structurally similar to the artificial preservatives, benzoic acid and 4-hydroxybenzoic acid, commonly added to foodstuff to prevent bacterial growth [13]. The lower activity detected by Russell et al. [12] could be due to the lower solubility in water of the ester compared to the corresponding acid. Interestingly, syringic acid has also been associated with adipogenesis inhibition and lipolysis promotion in 3T3-L1 adipocytes [14] and exhibited moderate to low antioxidant activity with IC_50_ values of 56–150 μg/mL and significant anti-urease activity with IC_50_ values in the range of 22–105 μg/mL when isolated from *Vincetoxicum stcoksii*, which is a perennial climbing leafy vine growing in Baluchistan [15].

Compound **3** was easily identified as the common lignan pinoresinol by detailed NMR investigation and comparison with the literature data [16]. Pinoresinol has been reported to exert antibacterial activity against *Pseudomonas aeruginosa* and *Bacillus subtilis* by inducing a disruptive action on bacterial cytoplasmic membrane [17].

The ^1^H NMR spectrum of compound **4** included aromatic signals at δ 7.07, d; 7.03, s; 6.91, d; signals of a two coupled *sp^2^* methines (*J* = 16 Hz, indicating *E* configuration) at 7.60 and 6.29, and signals of a long and unbranched alkyl chain including a methyl triplet at δ 0.88 and a deshielded oxymethylene at δ 4.19. The ^13^C NMR spectrum confirmed the presence of a trisubstituted phenyl unit including two oxygenated carbons and of an alkyl chain. Interpretation of HSQC and HMBC spectra clarified the presence of a ferulate unit esterified with a long chain (HMBC peak of H_2_-1′ with the ester carbonyl at δ 167.1). The HR-ESIMS spectrum of **4** was pivotal to deduce the C_14_ length of the long alkyl chain. Once we had completed the structural assignment of compound **4**, we became aware that this compound had been reported only once in the scientific literature, with the name erythrinassinate C, as a metabolite of the stems and root barks of *Erythrina sigmoidea* and *E. eriotricha* [18]. Interestingly, the genus *Erythrina* belongs to the family Fabaceae (Leguminosae) which is different from the family Plumbaginaceae of *Limonium* species.

It should be noted that synthetic ferulate esters structurally related to erythrinassinate C exerted a bacteriostatic and bactericidal effects on *E. coli* and *Listeria monocytogenes* with a mechanism based on the induction of cell elongation and destruction of the cell membrane with cell wall perforation [19]. The compounds have been suggested to be useful as food additives [19].

Finally, trace amounts of tyrosol palmitate (**5**) were also isolated and identified on the basis of a detailed NMR investigation and comparison with the literature data [20]. In particular, the ^1^H NMR spectrum of **5** showed the typical 2H-integrating and mutually coupled doublets of a *p*-disubstituted phenyl ring at δ 7.07 and 6.77, two mutually coupled methylenes (one of which is an oxymethylene, δ 4.24) and signals of a long saturated and unbranched fatty acid unit. Interpretation of 2D NMR spectra fully confirmed this assignment, while HR-ESIMS unambiguously indicated the structure of **5** as tyrosol palmitate. This is a very rare natural product reported only 3–4 times as a plant metabolite. The first paper reported its presence in *Ligustrum ovalifolium* (family Oleaceae) and therefore the compound was also called ligustrol A [20].

### 3.2. Antimicrobial Activity of the Secondary Metabolites Isolated from L. lopadusanum

The in vitro efficacy showed by the CH_2_Cl_2_ extract of *L. lopadusanum* in inhibiting the growth of *E. faecalis* appeared to be very interesting. *E. faecalis* is present in soil, water and food products and colonizes the human gastrointestinal tract. It has become one of the major opportunistic pathogens causing infections in both community and hospital settings. *E. faecalis* represents an important cause of complicated and uncomplicated urinary tract infections, hospital bacteremia, and wound infections, including diabetic foot ulcers, burns and surgical wounds [21,22]. It is one of the main causes of endocarditis, which mainly affects the elderly population, with a recently recorded dramatic increase of cases [23]. Enterococci can survive common antiseptics and disinfectants, favoring their persistence on inert hospital surfaces and objects [24]. This characteristic is probably one of the reasons for the involvement of *E. faecalis* in refractory apical periodontitis [25]. Furthermore, enterococci have intrinsic resistance to some antibiotics and can acquire resistance to other antimicrobials through mobile genetic elements [24]. Indeed, they have been included by the WHO among the growing causes of antibiotic-resistant infections that threaten public health [26]. Hence, there is a serious need to discover new promising drugs against these pathogenic bacteria. All the metabolites isolated from the CH_2_Cl_2_ organic extract, except compound **5**n which was obtained in too low amounts, were tested against *E. faecalis*, from 500 µg /mL (dilutions 1:2), and only compound **4** showed antibacterial (bacteriostatic) activity exhibiting a MIC at the concentration of 250 µg/mL (Table 2).

This is the first report of activity against enterococci of ferulate derivatives. However, synthetic ferulic acid esters with short chain fatty acids (FAC4, FAC6, FAC8, FAC10) have shown a good activity against *Escherichia coli* (Gram-), with the hexyl ferulate (FAC6) exerting excellent bacteriostatic and bactericidal effects [19]. It is interesting to notice that the *L. lopadusanum* extract showed poor activity against *E. coli*, indirectly suggesting the role of alkyl chain length in the modulation of the antibacterial activity.

## 4. Conclusions

In summary, our pytochemical investigation of the bioactive antibacterial extract of *L. lopadusanum* afforded five metabolites, all belonging to the class of phenolic derivatives, including simple phenols, ferulate and tyrosol esters and a lignan. Although phenolic compounds are widely occurring in the plant kingdom, we isolated two extremely rare metabolites, erythrinasinnate C and ligustrol A, the first isolated only once before and the second only a handful of times but, importantly, never reported before for their occurrence in the family Plumbaginaceae. The final results of our study candidate erythrinassinate C (**4**), a synthetically easily accessible molecule, for the development of a new drug against *E. faecalis*.

## Figures and Tables

**Figure 1 biomolecules-14-00134-f001:**
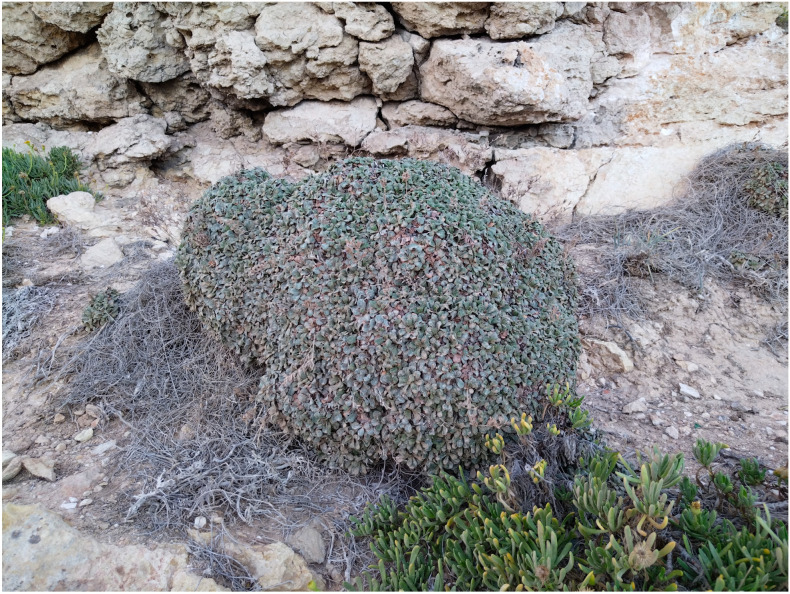
A bush of *Limonium lopadusanum* found at Capo Grecale, Lampedusa Island.

**Figure 2 biomolecules-14-00134-f002:**
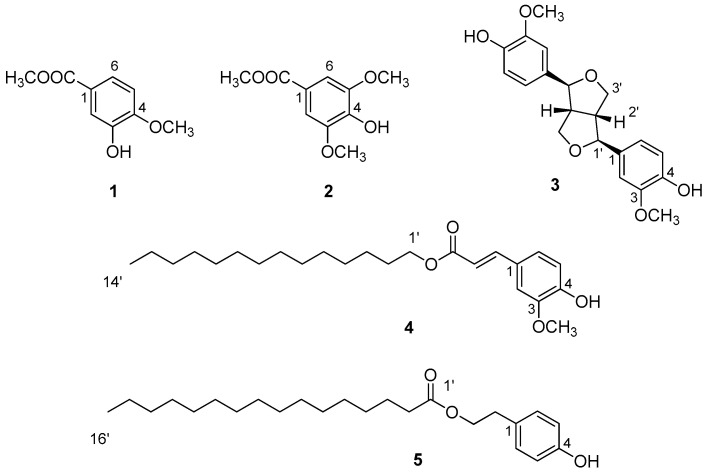
Chemical structures of metabolites isolated from the CH_2_Cl_2_ extract of *Limonium lopadusanum*.

**Table 1 biomolecules-14-00134-t001:** Antimicrobial activity of CH_2_Cl_2_ extract.

ATCC Strains	MIC (µg/mL)			
	*L. lopadusanus* (CH_2_Cl_2_ Extract)	Teicoplanin	Colistin	Voriconazole
*MRSA*	^a^	1	^b^	^b^
*E. faecalis*	500	0.5	^b^	^b^
*E. coli*	^a^	^b^	0.5	^b^
*A. baumannii*	^a^	^b^	0.5	^b^
*P. aeruginosa*	^a^	^b^	1	^b^
*C. albicans*	^a^	^b^	^b^	30

^a^ Not detected; ^b^ not Applicable.

**Table 2 biomolecules-14-00134-t002:** Antibacterial activity of secondary metabolites (**1**–**4**).

ATCC Strains	MIC (µg/mL)/MBC (µg/mL)				
	Compound 1	Compound 2	Compound 3	Compound 4	Teicoplanin
*E. faecalis*	ND	ND	ND	250/>500	0.5

ND—not detected.

## Data Availability

Data are contained within the article and Appendix A.

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
