# Peer review of "Antibacterial Metabolites Produced by Limonium lopadusanum, an Endemic Plant of Lampedusa Island"

_biomolecules, 2024, doi:10.3390/biom14010134_

Round 1

Reviewer 1 Report

Comments and Suggestions for Authors

Dear Authors,

one question has arisen regarding this sentence:

,,...Fresh plant materials (1.5 g) were extracted (1 × 3.0 L) with H2O/MeOH (1/1, v/v)...."     I would like to be sure: 1.5 g or 15 g? There is a huge difference in relation to the methanol volume. Or maybe 30 mL of methanol? This is only for my curiosity because such a ratio in an experimental work can exist.

Author Response

one question has arisen regarding this sentence:

,,...Fresh plant materials (1.5 g) were extracted (1 × 3.0 L) with H2O/MeOH (1/1, v/v)...."     I would like to be sure: 1.5 g or 15 g? There is a huge difference in relation to the methanol volume. Or maybe 30 mL of methanol? This is only for my curiosity because such a ratio in an experimental work can exist.

Answer: Sorry for the mistake the fresh plant material extracted was 1.5 kg. This has been corrected.

Reviewer 2 Report

Comments and Suggestions for Authors

Some comments:

The introduction section seems to be short. It would be better to develop more the part of the metabolites isolated from the same genre of the used plant as well as their potential biological activities.

The authors mention in the introduction section: Until now, there is no literature report on the secondary metabolites of L. lopadusanus and on their biological properties. I expect in extraction section they will use different solvents in order to investigate which solvent give a good biological activity and then go further in metabolites purification. However, the authors use only n-hexane and CH2Cl2 and no other solvent such as ethylacetate.  Kindly give explanation about the choice of used solvents.

In this present work, the authors purify and identify some metabolites from plant extracts and evaluate the antimicrobial activities of extracts as well as the pure compounds. In the results and discussion section I suggest to the authors to present the results of the antimicrobial activities in a table in which they collect all data related to results obtained from the plant organic extracts, pure compounds, controls (Teicoplanin, colistin and voriconazole), MIC, bacteria, fungi….It will help readers to more understand the results.

The authors use: * Teicoplanin (ranged from 0.06 to 4 μg/mL) and colistin (ranged from 0.5 to 32 μg/mL) as control conventional antibiotics for Gram-positive and Gram-negative, *voriconazole (ranged from 15 to 60 μg/mL) as conventional control for Candida strain, on the other hand they found that compound 4 showed antibacterial (bacteriostatic) activity exhibiting a MIC at the concentration of 250 μg/mL, which I think is high concentration compared to the control? Kindly give explanation regarding the huge difference in the concentration of the used control and the compound 4. Unfortunately, we don’t have the result obtained from the extract to compare the data.

In the extraction section the authors use 1.5 g fresh plant material and 3.0 L H2O/MeOH (1/1, v/v): Are you sure about the used volume???, and they obtain after extraction more than 600 mg oily residues, so the extraction yield is more than 40%??? Are you sure about that??

Scientific name should be written in Italic ex. 2.4. Purification of Metabolites from the L. lopadusanum Extracts

Comments on the Quality of English Language

Moderate editing of English language required

Author Response

The introduction section seems to be short. It would be better to develop more the part of the metabolites isolated from the same genre of the used plant as well as their potential biological activities.

ANSWER: A new paragraph was prepared accordingly and included in the revised version. Please, notice that three references were added.

The authors mention in the introduction section: Until now, there is no literature report on the secondary metabolites of L. lopadusanus and on their biological properties. I expect in extraction section they will use different solvents in order to investigate which solvent give a good biological activity and then go further in metabolites purification. However, the authors use only n-hexane and CH2Cl2 and no other solvent such as ethylacetate.  Kindly give explanation about the choice of used solvents.

ANSWER: in a preliminary experiment using different solvent as also chloroform and ethyl acetate. the metabolite profile was the same of that obtained with CH2Cl2 but the yield was slightly lower. This has been now stated in the discussion.

In this present work, the authors purify and identify some metabolites from plant extracts and evaluate the antimicrobial activities of extracts as well as the pure compounds. In the results and discussion section I suggest to the authors to present the results of the antimicrobial activities in a table in which they collect all data related to results obtained from the plant organic extracts, pure compounds, controls (Teicoplanin, colistin and voriconazole), MIC, bacteria, fungi….It will help readers to more understand the results.

ANSWER: Two Tables  (Table 1 and 2) were accordingly prepared and included in the revised version

The authors use: * Teicoplanin (ranged from 0.06 to 4 μg/mL) and colistin (ranged from 0.5 to 32 μg/mL) as control conventional antibiotics for Gram-positive and Gram-negative, *voriconazole (ranged from 15 to 60 μg/mL) as conventional control for Candida strain, on the other hand they found that compound 4 showed antibacterial (bacteriostatic) activity exhibiting a MIC at the concentration of 250 μg/mL, which I think is high concentration compared to the control? Kindly give explanation regarding the huge difference in the concentration of the used control and the compound 4. Unfortunately, we don’t have the result obtained from the extract to compare the data.

ANSWER: Control conventional antibiotics were used at the indicated ranges as ATCC strains were susceptible, except for Candida albicans. The search for unconventional antimicrobials is due to the spread of antimicrobial-resistant strains for which the MICs of conventional drugs are high and not achievable in vivo. In any case, we believe that an important result of our research is the identification of the compound 4 that is mainly responsible for the antimicrobial activity of the extract. In the Conclusion section we specified “The final results of our study candidate erythrinassinate C (4) as a potential, synthetically easily accessible, molecule for the development of a new drug against E. faecalis”. Indeed, our study will develop by introducing modifications to the scaffold of this molecule, in order to improve its antimicrobial activity.

In the extraction section the authors use 1.5 g fresh plant material and 3.0 L H2O/MeOH (1/1, v/v): Are you sure about the used volume???, and they obtain after extraction more than 600 mg oily residues, so the extraction yield is more than 40%??? Are you sure about that??

ANSWER: sorry for the mistake the fresh plant material extracted was 1.5 kg. This has been corrected

Scientific name should be written in Italic ex. 2.4. Purification of Metabolites from the L. lopadusanum Extracts

ANSWER: Revised